# On Single Source Robustness in Deep Fusion Models

**Taewan Kim**[*]
The University of Texas at Austin
Austin, TX
twankim@utexas.edu

**Joydeep Ghosh**
The University of Texas at Austin
Austin, TX
jghosh@utexas.edu

## Abstract

Algorithms that fuse multiple input sources benefit from both complementary and shared information. Shared information may provide robustness against faulty or noisy inputs, which is indispensable for safety-critical applications like self-driving cars. We investigate learning fusion algorithms that are robust against noise added to a single source. We first demonstrate that robustness against single source noise is not guaranteed in a linear fusion model. Motivated by this discovery, two possible approaches are proposed to increase robustness: a carefully designed loss with corresponding training algorithms for deep fusion models, and a simple convolutional fusion layer that has a structural advantage in dealing with noise. Experimental results show that both training algorithms and our fusion layer make a deep fusion-based 3D object detector robust against noise applied to a single source, while preserving the original performance on clean data.

## 1 Introduction

Deep learning models have accomplished superior performance in several machine learning problems [26] including object recognition [24, 40, 42, 15, 18], object detection [37, 16, 7, 36, 30, 35] and speech recognition [17, 14, 38, 5, 2, 4], which use either visual or audio sources. One natural way of improving a model's performance is to make use of multiple input sources relevant to a given task so that enough information can be extracted to build strong features. Therefore, deep fusion models have recently attracted considerable attention for autonomous driving [21, 3, 33, 25], medical imaging [23, 48, 39, 29], and audio-visual speech recognition [19, 32, 41, 6].

Two benefits are expected when fusion-based learning models are selected for a given problem. First, given adequate data, more information from multiple sources can enrich the model's feature space to achieve higher prediction performance, especially, when different input sources provide *complementary information* to the model. This expectation coincides with a simple information theoretic fact: if we have multiple input sources $X_1, \cdots, X_{n_s}$ and a target variable $Y$, mutual information $I(;)$ obeys $I(Y; X_1, \cdots, X_{n_s}) \geq I(Y; X_i)$ $(\forall i \in [n_s])$.

The second expected advantage is increased robustness against single source faults, which is the primary concern of our work. An underlying intuition comes from the fact that different sources may have *shared information* so one sensor can partially compensate for others. This type of robustness is critical in real-world fusion models, because each source may be exposed to different types of corruption but not at the same time. For example, self-driving cars using an RGB camera and ranging sensors like LIDAR and radar are exposed to single source corruption. LIDARs and radars work fine at night whereas RGB cameras do not. Also, each source used in the model may have its own sensing device, and hence not necessarily be corrupted by some physical attack simultaneously with others. It would be ideal if the structure of machine learning based fusion models and shared information could compensate for the corruption and automatically guarantee robustness without additional steps.

---

[*]This work was done when Taewan Kim was at the University of Texas at Austin, prior to joining Amazon.

This paper shows that a fusion model needs a supplementary strategy and a specialized structure to avoid vulnerability to noise or corruption on a single source. Our contributions are as follows:

- We show that a fusion model learned with a standard robustness is not guaranteed to provide robustness against noise on a single source. Inspired by the analysis, a novel loss is proposed to achieve the desired robustness (Section 3).
- Two efficient training algorithms for minimizing our loss in deep fusion models are devised to ensure robustness without impacting performance on clean data (Section 4.1).
- We introduce a simple but an effective fusion layer which naturally reduces error by applying ensembling to latent convolutional features (Section 4.2).

We apply our loss and the fusion layer to a complex deep fusion-based 3D object detector used in autonomous driving for further investigation in practice. Note that our findings can be easily generalized to other applications exhibiting intermittent defects in a subset of input sources, e.g., robustness given $k$ of $n_s$ corrupted sources, and single source robustness should be studied in depth prior to more general cases.

## 2   Related Works

Deep fusion models have been actively studied in object detection for autonomous vehicles. There exist two major streams classified according to their algorithmic structures: two-stage detectors with R-CNN (Region-based Convolutional Neural Networks) technique [12, 11, 37, 7, 16], and single stage detectors for faster inference speed [36, 35, 30].

Earlier deep fusion models extended Fast R-CNN [11] to provide better quality of region proposals from multiple sources [21, 1]. With a high-resolution LIDAR, point cloud was used as a major source of the region proposal stage before the fusion step [8], whereas F-PointNet [33] used it for validating 2D proposals from RGB images and predicting 3D shape and location within the visual frustum. MV3D [3] extended the idea of region proposal network (RPN) [37] by generating proposals from RGB image, and LIDAR's front view and BEV (bird's eye view) maps. Recent works tried to remove region proposal stages for faster inference and directly fused LIDAR's front view depth image [22] or BEV image [47] with RGB images. ContFuse [27] utilizes both RGB and LIDAR's BEV images with a new continuous fusion scheme, which is further improved in MMF [28] by handling multiple tasks at once. Our experimental results are based on AVOD [25], a recent open-sourced 3D object detector that generates region proposals from RPN using RGB and LIDAR's BEV images.

Compared to the active efforts in accomplishing higher performance on clean data, very few works have focused on robust learning methods in multi-source settings to the best of our knowledge. Adaptive fusion methods using gating networks weight the importance of each source automatically [31, 46], but these works lack in-depth studies of the robustness against single source faults. A recent work proposed a gated fusion at the feature level and applied data augmentation techniques with randomly chosen corruption methods [20]. In contrast, our training algorithms are surrogate minimization schemes for the proposed loss function, which is grounded from the analyses on underlying weakness of fusion methods. Also the fusion layer proposed in this paper focuses more on how to mix convolutional feature maps *channel-wise* with simple trainable procedures. For extensive literature reviews, please refer to the recent survey papers about deep multi-modal learning methods in general [34] and for autonomous driving [9].

## 3   Single Source Robustness of Fusion Models

### 3.1   Regression on linear fusion data

To show the vulnerability of naive fusion models, we introduce a simple data model and a fusion algorithm. Suppose $y$ is a linear function consisting of three different inherent (latent) components $z_i \in \mathbb{R}^{d_i}$ ($i \in \{1, 2, 3\}$). There are two input sources, $x_1$ and $x_2$. Here $\psi$'s are unknown functions.

$$y = \sum_{i=1}^{3} \beta_i^T z_i, \ \text{ where } \ z_1 = \psi_1(x_1), \ z_2 = \psi_2(x_2), \ z_3 = \psi_{3,1}(x_1) = \psi_{3,2}(x_2) \qquad (1)$$

Our simple data model simulates a target variable $y$ relevant to two different sources, where each source has its own special information $z_1$ and $z_2$ and a shared one $z_3$. For example, if two sources are obtained from an RGB camera and a LIDAR sensor, one can imagine that any features related to objectness are captured in $z_3$ whereas colors and depth information may be located in $z_1$ and $z_2$, respectively. Our objective is to build a regression model by effectively incorporating information from the sources $(x_1, x_2)$ to predict the target variable $y$.

Now, consider a fairly simple setting $x_1 = [z_1; z_3] \in \mathbb{R}^{d_1+d_3}$ and $x_2 = [z_2; z_3] \in \mathbb{R}^{d_2+d_3}$, where $(\psi_1, \psi_2, \psi_{3,1}, \psi_{3,2})$ can be defined accordingly to satisfy (1). A straightforward fusion approach is to stack the sources, i.e. $x = [x_1; x_2] \in \mathbb{R}^{d_1+d_2+2d_3}$, and learn a linear model. Then, it is easy to show that there exists a feasible *error-free model* for noise-free data:

$$f_{\text{direct}}(x_1, x_2) = h_1^T x_1 + h_2^T x_2 = (\beta_1^T z_1 + g_1^T z_3) + (\beta_2^T z_2 + g_2^T z_3), \ s.t. \ g_1 + g_2 = \beta_3 \quad (2)$$

where $h_1 = [\beta_1; g_1], h_2 = [\beta_2; g_2]$. Parameter vectors responsible for the shared information $z_3$ are denoted by $g_1$ and $g_2$.[2]

**Unbalanced robustness (Motivation)**   Suppose the true parameters of data are scalar values, i.e. $\beta_i = c_i \in \mathbb{R}$ and influence of the complementary information is relatively small, $c_1 \approx c_2$ and $c_3 \gg c_1$. Assume that the obtained error-free solution's parameters for $z_3$ are unbalanced, i.e. $g_1 = \Delta$ and $g_2 = c_3 - \Delta$ with some weight parameter $\Delta \ll c_3$, so that $g_1$ gives a negligible contribution. Then add single source corruption $\delta_1 = [\epsilon_1; \epsilon_3]$ and $\delta_2 = [\epsilon_2; \epsilon_3]$ and compute absolute difference between the true data $y$ and the prediction from the corrupted data:

$$|y - f_{\text{direct}}(x_1 + \delta_1, x_2)| = |c_1 \epsilon_1 + \Delta \epsilon_3|, \qquad |y - f_{\text{direct}}(x_1, x_2 + \delta_2)| = |c_2 \epsilon_2 + (c_3 - \Delta)\epsilon_3|$$

In this case, adding noise to the source $x_2$ will give significant corruption to the prediction while $x_1$ is relatively robust because $|(c_3 - \Delta)\epsilon_3| \gg |\Delta \epsilon_3|$ for any noise $\epsilon_3$ affecting $z_3$. This simple example illustrates that additional training strategies or components are indispensable to achieve robust fusion model working even if one of the sources is disturbed. The next section introduces a novel loss for a balanced robustness against a fault in a single source.

## 3.2   Robust learning for single source noise

Fusion methods are not guaranteed to provide robustness against faults in a single source without additional supervision. Also, we demonstrate that naive regularization or robust learning methods are not sufficient for the robustness later in this section. Therefore, a supplementary constraint or strategy needs to be considered in training which can correctly guide learning parameters for the desired robustness.

One essential requirement of fusion models is showing *balanced performance* regardless of corruption added to any source. If the model is significantly vulnerable to corruption in one source, this model becomes untrustworthy and we need to balance the degradation levels of different input sources' faults. For example, suppose there is a model robust against noise in RGB channels but shows huge degradation in performance for any fault of LIDAR. Then the overall system should be considered untrustworthy, because there exist certain corruption or environments which can consistently fool the model. Our loss, MAXSSN (Maximum Single Source Noise), for such robustness is introduced to handle this issue and further analyses are provided under the linear fusion data model explained in Section 3.1. This loss makes the model focus more on corruption of a single source, SSN, rather than focusing on noise added to all the sources at once, ASN.

**Definition 1.** *For multiple sources $x_1, \cdots, x_{n_s}$ and a target variable $y$, denote a predefined loss function by $\mathcal{L}$. If each source $x_i$ is perturbed with some additive noise $\epsilon_i$ for $i \in [n_s]$, MAXSSN loss for a model $f$ is defined as follows:*

$$\mathcal{L}_{\text{MAXSSN}}(f, \epsilon) \triangleq \max_i \{\mathcal{L}(y, f(x_1, \cdots, x_{i-1}, x_i + \epsilon_i, x_{i+1}, \cdots, x_{n_s}))\}_{i=1}^{n_s}$$

Another key principle in our robust training is to *retain the model's performance on clean data*. Although techniques like data augmentation help improving a model's generalization error in general, learning a model robust against certain types perturbation including adversarial attacks may harm the model's accuracy on non-corrupt data [43]. Deterioration in the model's ability on normal data is an unwanted side effect, and hence our approach aims to avoid this.

**Random noise**  To investigate the importance of our MAXSSN loss, we revisit the linear fusion data model with the optimal direct fusion model $f_{\text{direct}}$ of the regression problem introduced in Section 3.1. Suppose the objective is to find a model with robustness against single source noises, while preserving error-free performance, i.e., unchanged loss under clean data. For the noise model, consider $\epsilon = [\delta_1; \delta_2]$ where $\delta_1 = [\epsilon_1; \epsilon_3]$ and $\delta_2 = [\epsilon_2; \epsilon_4]$, which satisfy $\mathbb{E}[\epsilon_i] = 0$, $Var(\epsilon_i) = \sigma^2 I$, and $\mathbb{E}[\epsilon_i \epsilon_j^T] = 0$ for $i \neq j$. Note that noises added to the shared information, $\epsilon_3$ and $\epsilon_4$, are not identical, which resembles direct perturbation to the input sources in practice. For example, noise directly affecting a camera lens does not need to perturb other sources.

**Optimal fusion model for MAXSSN**  The robust linear fusion model $f(x_1, x_2) = (w_1^T z_1 + g_1^T z_3) + (w_2^T z_1 + g_2^T z_3)$ is found by minimizing $\mathcal{L}_{\text{MAXSSN}}(f, \epsilon)$ over parameters $w_1, w_2, g_1$ and $g_2$. As shown in the previous section, any $f_{\text{direct}}$ satisfying $w_1 = \beta_1, w_2 = \beta_2$ and $g_1 + g_2 = \beta_3$ should achieve zero-error. Therefore, overall optimization problem can be reduced to the following one:

$$\min_{g_1, g_2} \max \left\{ \mathcal{L}\left(y, f_{\text{direct}}(x_1 + \delta_1, x_2)\right), \mathcal{L}\left(y, f_{\text{direct}}(x_1, x_2 + \delta_2)\right) \right\} \quad s.t. \ g_1 + g_2 = \beta_3 \quad (3)$$

If we use a standard expected squared loss $\mathcal{L}(y, f(x_1, x_2)) = \mathbb{E}[(y - f(x_1, x_2))^2]$ and solve the optimization problem, the following solution $\mathcal{L}^*_{\text{MAXSSN}}$ with corresponding parameters $g_1^*, g_2^*$ can be obtained, and there exist three cases based on the relative sizes of $||\beta_i||_2$'s.

$$(\mathcal{L}^*_{\text{MAXSSN}}, \ g_1^*, \ g_2^*) = \begin{cases} \left(\sigma^2 ||\beta_2||_2^2, \ \beta_3, \ 0\right) & \text{if } ||\beta_1||_2^2 + ||\beta_3||_2^2 \leq ||\beta_2||_2^2 \\ \left(\sigma^2 ||\beta_1||_2^2, \ 0, \ \beta_3\right) & \text{if } ||\beta_2||_2^2 + ||\beta_3||_2^2 \leq ||\beta_1||_2^2 \\ \left(\sigma^2 \left(\frac{||\beta_1||_2^2 + ||\beta_2||_2^2}{2} + \frac{||\beta_3||_2^2}{4} + \frac{(||\beta_2||_2^2 - ||\beta_1||_2^2)^2}{4||\beta_3||_2^2}\right), \right. \\ \left. \frac{1}{2}\left(1 + \frac{||\beta_2||_2^2 - ||\beta_1||_2^2}{||\beta_3||_2^2}\right), \ \frac{1}{2}\left(1 - \frac{||\beta_2||_2^2 - ||\beta_1||_2^2}{||\beta_3||_2^2}\right)\right) & \text{otherwise} \end{cases} \quad (4)$$

The three cases reflect the relative influence of each weight vector for $z_i$. For instance, if $z_2$ has larger importance compared to the rest in generating $y$, the optimal way of balancing the effect of noise over $z_3$ is to remove all the influence of $z_2$ in $x_2$ by setting $g_2 = 0$. When neither of $z_1$ nor $z_2$ dominates the importance, i.e. $\left| \frac{||\beta_2||_2^2 - ||\beta_1||_2^2}{||\beta_3||_2^2} \right| < 1$, the optimal solution tries to make $\mathcal{L}\left(y, f_{\text{direct}}(x_1 + \delta_1, x_2)\right) = \mathcal{L}\left(y, f_{\text{direct}}(x_1, x_2 + \delta_2)\right)$.

**Comparison with the standard robust fusion model**  Minimizing loss with noise added to a model's input is a standard process in robust learning. The same strategy can be applied to learn fusion models by considering all sources as a single combined source, then add noise to all the sources at once. However, this simple strategy cannot achieve low error in terms of the single source robustness. The optimal solution to $\min_{g_1, g_2} \mathbb{E}[(y - f_{\text{direct}}(x_1 + \delta_1, x_2 + \delta_2))^2]$, a least squares solution, is achieved when $g_1 = g_2 = \frac{\beta_3}{2}$. The corresponding MAXSSN loss can be evaluated as $\mathcal{L}'_{\text{MAXSSN}} = \sigma^2 \max \left\{ ||\beta_1||_2^2 + \frac{1}{4}||\beta_3||_2^2, ||\beta_2||_2^2 + \frac{1}{4}||\beta_3||_2^2 \right\}$. A nontrivial gap exists between $\mathcal{L}_{\text{MAXSSN}}$ and $\mathcal{L}'_{\text{MAXSSN}}$, which is directly proportional to the data model's inherent characteristics:

$$\mathcal{L}'_{\text{MAXSSN}} - \mathcal{L}^*_{\text{MAXSSN}} \geq \begin{cases} \frac{1}{4}||\beta_3||_2^2 & \text{if } \left| \frac{||\beta_2||_2^2 - ||\beta_1||_2^2}{||\beta_3||_2^2} \right| \geq 1 \\ \frac{1}{4} \left| ||\beta_2||_2^2 - ||\beta_1||_2^2 \right| & \text{otherwise} \end{cases} \quad (5)$$

If either $z_1$ or $z_2$ has more influence on the target value $y$ than the other components, single source robustness of the model trained by MAXSSN loss is better than the fusion model for the general noise robustness with an amount proportional to the influence of shared feature $z_3$. Otherwise, the gap's lower bound is proportional to the difference in complementary information, $|||\beta_2||_2^2 - ||\beta_1||_2^2|/4$.

**Remark 1.**  In linear systems such as the one studied above, having redundant information in the feature space is similar to multicollinearity in statistics. In this case, feature selection methods usually try to remove such redundancy. However, this redundant or shared information helps preventing degradation of the fusion model when a subset of the input sources are corrupted.

**Remark 2.**  Similar analyses and a loss definition against *adversarial attacks* [13] are provided in appendix A.2.

## 4 Robust Deep Fusion Models

In simple linear settings, our analyses illustrate that using MAXSSN loss can effectively minimize the degradation of a fusion model's performance. This suggests a training strategy for complex deep fusion models to be equipped with robustness against single source faults. A principal factor considered in designing a common framework for our algorithms is the preservation of model's performance on clean data while minimizing a loss for defending corruption. Therefore, our training algorithms use *data augmentation* to encounter both clean and corrupted data. The second way of achieving robustness is to take advantage of the fusion method's structure. A simple but effective method of mixing convolutional features coming from different input sources is introduced later in this section.

### 4.1 Robust training algorithms for single source noise

In the previous section, we solve problem (3) by optimizing over flexible parameters $g_1$ and $g_2$. If the parts of input sources contributing to $z_3$ are known, then indeed we can achieve this goal. In practice however, it is difficult to know which parts of an input source (or latent representation) are related to shared information and which parameters are flexible. Therefore, our common training framework alternately provides *clean samples* and *corrupted samples* per iteration to preserve the original performance of the model on uncontaminated data.[3] On top of this strategy, one standard robust training scheme and two algorithms for minimizing MAXSSN loss are introduced for handling robustness against noise in different sources.

**Standard robust training method**   A standard robust training algorithm can be developed by considering all $n_s$ sources as a single combined source. Given noise generating functions $\varphi_i(\cdot)$ ($i \in [n_s]$), the algorithm generates and adds corruption to all the sensors at once. Then the corresponding loss can be computed to update parameters using back-propagation. This algorithm is denoted by TRAINASN and tested in experiments to investigate whether the procedure is also able to cover robustness against single source noise.

---

**Algorithm 1** TRAINSSN

> **for** $i_{\text{iter}} = 1$ to $m$ **do**
>  Sample $(y, \{x_i\}_{i=1}^{n_s})$
>  **if** $i_{\text{iter}} \equiv 1 \pmod 2$ **then**
>   **for** $j = 1$ to $n_s$ **do**
>    Generate noise $\epsilon_j = \varphi_j(x_j)$
>    $\hat{\mathcal{L}}_j^{(i_{\text{iter}})} \leftarrow \mathcal{L}(y, f(\{x_j + \epsilon_j, x_{-j}\}))$
>   **end for**
>   $\mathcal{L}^{(i_{\text{iter}})} \leftarrow \max_j \hat{\mathcal{L}}_j^{(i_{\text{iter}})}$
>  **else**
>   $\mathcal{L}^{(i_{\text{iter}})} \leftarrow \mathcal{L}(y, f(\{x_i\}_{i=1}^{n_s}))$
>  **end if**
>  Update $f$ using $\nabla \mathcal{L}^{(i_{\text{iter}})}$
> **end for**

**Algorithm 2** TRAINSSNALT

> **for** $i_{\text{iter}} = 1$ to $m$ **do**
>  Sample $(y, \{x_i\}_{i=1}^{n_s})$
>  **if** $i_{\text{iter}} \equiv 1 \pmod 2$ **then**
>   $j \leftarrow (\lfloor i_{\text{iter}}/2 \rfloor \bmod n_s) + 1$
>   Generate noise $\epsilon_j = \varphi_j(x_j)$
>   $\mathcal{L}^{(i_{\text{iter}})} \leftarrow \mathcal{L}(y, f(\{x_j + \epsilon_j, x_{-j}\}))$
>  **else**
>   $\mathcal{L}^{(i_{\text{iter}})} \leftarrow \mathcal{L}(y, f(\{x_i\}_{i=1}^{n_s}))$
>  **end if**
>  Update $f$ using $\nabla \mathcal{L}^{(i_{\text{iter}})}$
> **end for**

---

**Minimization of MAXSSN loss**   Minimization of the MAXSSN loss requires $n_s$ (number of input sources) forward-propagations within one iteration. Each propagation needs a different set of corrupted samples generated by adding single source noise to the fixed clean mini-batch of data. There are two possible approaches to compute gradients properly from these multiple passes. First, we can run back-propagation $n_s$ times to save the gradients temporarily without updating any parameters, then the saved gradients with the maximum loss is used for updating parameters.

However, this process requires not only $n_s$ forward and backward passes but also large memory usage proportional to $n_s$ for saving the gradients. Another reasonable approach is to run $n_s$ forward passes to find the maximum loss and compute gradients by going back to the corresponding set of corrupted samples. Algorithm 1 adopts this idea for its efficiency, $n_s + 1$ forward passes and one back-propagation. A faster version of the algorithm, TRAINSSNALT, is also considered since multiple forward passes may take longer as the number of sources increases. This algorithm ignores the maximum loss and alternately augments corrupted data. By a slight abuse of notation, symbols used in our algorithms also represent the iteration steps with the size of mini-batches greater than one. Also, $f(x_1, \cdots, x_{j-1}, x_j + \epsilon_j, x_{j+1}, \cdots, x_{n_s})$ is shortened to $f(\{x_j + \epsilon_j, x_{-j}\})$ in the algorithms.

## 4.2 Feature fusion methods

Fusion of features extracted from multiple input sources can be done in various ways [3]. One of the popular methods is to fuse via an element-wise mean operation [25], but this assumes that each feature must have a same shape, i.e., width, height, and number of channels for a 3D feature. An element-wise mean can be also viewed as averaging channels from different 3D features, and it has an underlying assumption that the channels of each feature should share same information regardless of the input source origin. Therefore, the risk of becoming vulnerable to single source corruption may increase with this simple mean fusion method.

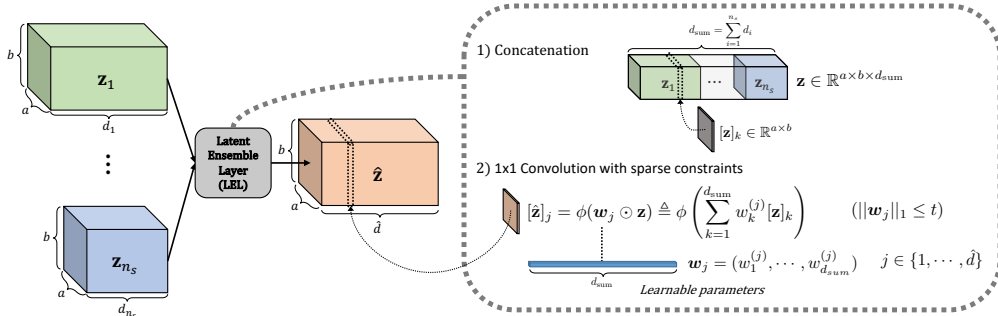

Figure 1: Latent ensemble layer (LEL)

Our fusion method, latent ensemble layer (LEL), is devised for three objectives: (i) maintaining the known advantage—error reduction—of ensemble methods [45, 44], (ii) admitting source-specific features to survive even after the fusion procedure, and (iii) allowing each source to provide a different number of channels. The proposed layer learns parameters so that channels of the 3D features from the different sources can be selectively mixed. Sparse constraints are introduced to let the training procedure find good subsets of channels to be fused across the $n_s$ feature maps. For example, mixing the $i^{\text{th}}$ channel of the convolutional feature from an RGB image with the $j^{\text{th}}$ and $k^{\text{th}}$ channels of the LIDAR's latent feature is possible in our LEL, whereas in an element-wise mean layer the $i^{\text{th}}$ latent channel from RGB is only mixed with the other sources' $i^{\text{th}}$ channels.

In practice, this layer can be easily constructed by using $1 \times 1$ convolutions with the ReLU activation and $\ell_1$ constraints. We also apply an activation function to supplement a semi-adaptive behavior to the fusion procedure. Depth of the output channel is set to $\hat{d} = \max_i\{d_i\}$ and we set the hyper-parameter for $\ell_1$ constraint as 0.01 in the experiments. Definition 2 explains the details of our LEL, and Figure 1 visualizes the overall process.

**Definition 2** (Latent ensemble layer). *Suppose we have $n_s$ convolutional features $\mathbf{z}_i \in \mathbb{R}^{a \times b \times d_i}$ from different input sources ($i \in [n_s]$), which can be stacked as $\mathbf{z} = (\mathbf{z}_1, \cdots, \mathbf{z}_m) \in \mathbb{R}^{a \times b \times d_{sum}}$ ($d_{sum} = \sum_{i=1}^m d_i$). The $k^{th}$ channel of the stacked feature is denoted by $[\mathbf{z}]_k \in \mathbb{R}^{a \times b}$. Let $\boldsymbol{w}_j = (w_1^{(j)}, \cdots, w_{d_{sum}}^{(j)})$ be a $d_{sum}$-dimensional weight vector to mix $\mathbf{z}_i$'s in channel-wise fashion. Then LEL outputs $\hat{\mathbf{z}} \in \mathbb{R}^{a \times b \times \hat{d}}$ where each channel is computed as $[\hat{\mathbf{z}}]_j = \phi(\boldsymbol{w}_j \odot \mathbf{z}) \triangleq \phi\left(\sum_{k=1}^{d_{sum}} w_k^{(j)}[\mathbf{z}]_k\right)$, with some activation function $\phi$ and sparse constraints $||\boldsymbol{w}_j||_0 \leq t$ for all $j \in \{1, \cdots, \hat{d}\}$.*

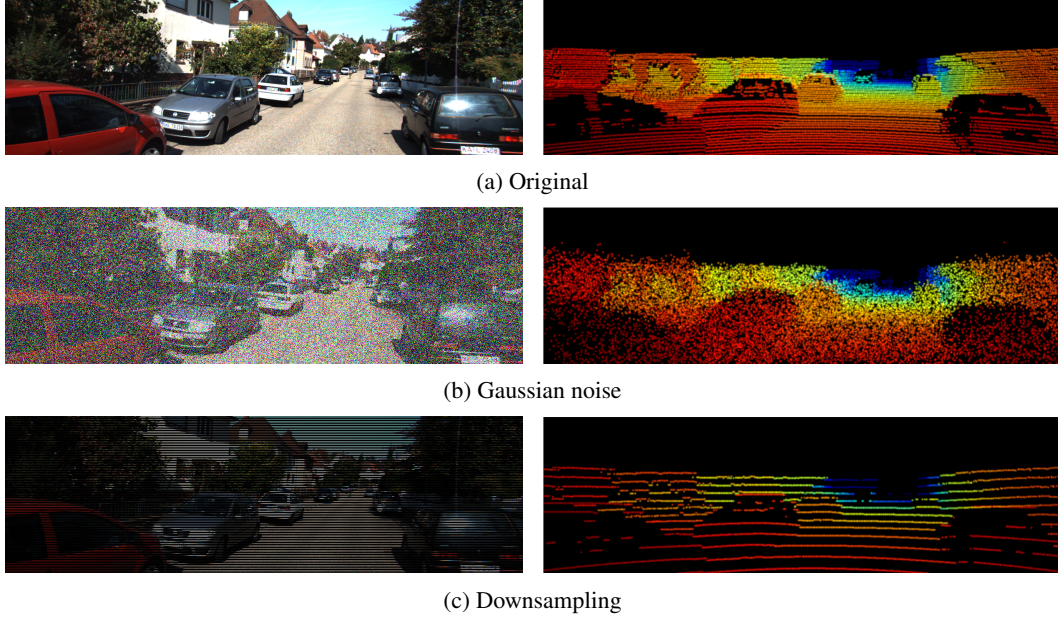

(a) Original

(b) Gaussian noise

(c) Downsampling

Figure 2: Visualization of corrupted samples, *(left)* RGB images *(right)* LIDAR point clouds. The point clouds are projected onto the 2D image plane for easier visual comparison.

# 5   Experimental Results

We test our algorithms and the LEL fusion method on 3D and BEV object detection tasks using the car class of the KITTI dataset [10]. 3D detection is both an important problem in self-driving cars and one where multiple sensors can contribute fruitfully by providing both complementary and shared information. In contrast, models for 2D object detection heavily rely on RGB data, which typically dominates other modalities. As our experiments include random generation of corruption, each task is evaluated 5 times to compare average scores (reported with 95% confidence intervals), and thus a validation set is used for ease of manipulating data and repetitive evaluation. We follow the split of Ku et al. [25], 3712 and 3769 frames for training and validation sets, respectively. Results are reported based on three difficulty levels defined by KITTI (easy, medium, hard) and a standard metric Average Precision (AP) is used. A recent open-sourced 3D object detector AVOD [25] with a feature pyramid network is selected as a baseline algorithm.

Four different algorithms are compared: AVOD trained on (i) clean data, (ii) data augmented with ASN samples (TRAINASN), (iii) SSN augmented data with direct MAXSSN loss minimization (TRAINSSN), and (iv) SSN augmented data (TRAINSSNALT). The AVOD architecture is varied to use either element-wise mean fusion layers or our LELs. We follow the original training setups of AVOD, e.g., 120k iterations using an ADAM optimizer with an initial learning rate of 0.0001.[4]

**Corruption methods**   *Gaussian noise* generated i.i.d. with $\mathcal{N}(0, \sigma_{\text{Gaussian}}^2)$ is directly added to the pixel value of an image $(r, g, b)$ and the coordinate value of a LIDAR's point $(x, y, z)$. $\sigma_{\text{Gaussian}}$ is set to $0.75\tau$ experimentally with $\tau_{\text{RGB}} = 255$ and $\tau_{\text{LIDAR}} = 0.2$. The second method *downsampling* selects only 16 out of 64 lasers of LIDAR data. To match this effect, 3 out of 4 horizontal lines of an RGB image are deleted. Effects of corruption on each input source are visualized in Figure 2, where the color of a 2D LIDAR image represents a distance from the sensor. Although our analyses in Section 3.2 assume the noise variances to be identical, it is nontrivial to set equal noise levels for different modalities in practice, e.g., RGB pixels vs. points in a 3D space. Nevertheless, an underlying objective of our MAXSSN loss, balancing the degradation rates of different input sources' faults, does not depend on the choice of noise types or levels.

**Evaluation metrics for single source robustness**    To assess the robustness against single source noise, a new metric minAP is introduced. The AP score is evaluated on the dataset with a single corrupted input source, then after going over all $n_s$ sources, minAP reports the lowest score among the $n_s$ AP scores. Our second metric maxDiffAP computes the maximum absolute difference among the scores, which measures the balance of different input sources' single source robustness; low value of maxDiffAP means the well-balanced robustness.

Table 1: Car detection (3D/BEV) performance of AVOD with *element-wise mean* fusion layers and *latent ensemble layers (LEL)* against *Gaussian* SSN on the KITTI validation set.

| (Data) Train Algo. | Easy | Moderate | Hard | Easy | Moderate | Hard |
|---|---|---|---|---|---|---|
| | | ***Fusion method***: *Mean* | | | | |
| (Clean Data) | | $AP_{3D}(\%)$ | | | $AP_{BEV}(\%)$ | |
| AVOD [25] | **76.41** | **72.74** | **66.86** | 89.33 | 86.49 | **79.44** |
| +TRAINASN | 75.96 | 66.68 | 65.97 | 88.63 | 79.45 | 78.79 |
| +TRAINSSN | 76.28 | 67.10 | 66.51 | 88.86 | 79.60 | 79.11 |
| +TRAINSSNALT | **77.46** | 67.61 | 66.06 | **89.68** | **86.71** | 79.41 |
| (Gaussian SSN) | | $\min AP_{3D}(\%)$ | | | $\min AP_{BEV}(\%)$ | |
| AVOD [25] | 47.41±0.28 | 41.84±0.17 | 36.47±0.16 | 65.63±0.28 | 58.02±0.23 | 50.43±0.14 |
| +TRAINASN | 61.53±0.57 | 52.72±0.08 | 47.25±0.13 | 87.71±0.14 | 78.37±0.06 | 77.85±0.08 |
| +TRAINSSN | 71.65±0.31 | **62.14±0.08** | **56.78±0.12** | 88.21±0.08 | 78.90±0.09 | **77.92±0.11** |
| +TRAINSSNALT | **71.66±0.48** | 57.61±0.12 | 55.90±0.11 | **89.42±0.04** | **79.56±0.06** | **77.92±0.05** |
| (Gaussian SSN) | | $\max \mathrm{Diff} AP_{3D}(\%)$ | | | $\max \mathrm{Diff} AP_{BEV}(\%)$ | |
| AVOD [25] | 26.70±0.52 | 22.42±0.29 | 20.92±0.25 | 22.27±0.41 | 20.76±0.33 | 20.09±0.20 |
| +TRAINASN | 14.48±0.82 | 12.72±0.33 | 11.18±0.27 | 0.88±0.22 | 0.48±0.13 | 0.28±0.12 |
| +TRAINSSN | **3.71±0.46** | **3.42±0.25** | 7.50±0.25 | 0.36±0.17 | **0.04±0.15** | 0.71±0.17 |
| +TRAINSSNALT | 5.55±0.81 | 8.73±0.32 | **2.91±0.22** | **0.09±0.14** | 0.13±0.11 | **0.18±0.11** |
| | | ***Fusion method***: *Latent Ensemble Layer* | | | | |
| (Clean Data) | | $AP_{3D}(\%)$ | | | $AP_{BEV}(\%)$ | |
| AVOD [25] | **77.79** | **67.69** | **66.31** | **88.90** | **85.64** | **78.86** |
| +TRAINASN | 75.00 | 64.75 | 58.28 | 88.30 | 78.60 | 77.23 |
| +TRAINSSN | 74.25 | 65.00 | 63.83 | 87.88 | 78.84 | 77.66 |
| +TRAINSSNALT | 76.04 | 66.42 | 64.41 | 88.80 | 79.53 | 78.53 |
| (Gaussian SSN) | | $\min AP_{3D}(\%)$ | | | $\min AP_{BEV}(\%)$ | |
| AVOD [25] | 61.97±0.55 | 53.95±0.42 | 47.24±0.27 | 79.44±0.09 | 72.46±3.14 | 68.25±0.06 |
| +TRAINASN | **74.24±0.38** | 58.25±0.16 | **56.13±0.10** | 88.10±0.26 | **78.19±0.13** | 70.42±0.07 |
| +TRAINSSN | 68.16±0.88 | **60.39±0.38** | 56.04±0.28 | **88.12±0.16** | 78.17±0.06 | 70.21±0.05 |
| +TRAINSSNALT | 68.63±0.40 | 55.48±0.16 | 54.42±0.17 | 86.51±0.46 | 76.85±0.11 | **71.95±2.72** |

Table 2: Car detection (3D/BEV) performance of AVOD with *latent ensemble layers (LEL)* against *downsampling* SSN on the KITTI validation set.

| (Data) Train Algo. | Easy | Moderate | Hard | Easy | Moderate | Hard |
|---|---|---|---|---|---|---|
| (Clean Data) | | $AP_{3D}(\%)$ | | | $AP_{BEV}(\%)$ | |
| AVOD [25] | **77.79** | **67.69** | **66.31** | 88.90 | **85.64** | **78.86** |
| +TRAINASN | 71.74 | 61.78 | 60.26 | 87.29 | 77.08 | 75.89 |
| +TRAINSSN | 75.54 | 66.26 | 63.72 | 88.07 | 79.18 | 78.03 |
| +TRAINSSNALT | 76.22 | 66.05 | 63.87 | **89.00** | 79.65 | 78.03 |
| (Downsample SSN) | | $\min AP_{3D}(\%)$ | | | $\min AP_{BEV}(\%)$ | |
| AVOD [25] | 61.70 | 51.66 | 46.17 | 86.08 | 69.99 | 61.55 |
| +TRAINASN | 65.74 | 53.49 | 51.35 | 82.27 | 67.88 | 65.79 |
| +TRAINSSN | **73.33** | **57.85** | **54.91** | **86.61** | **76.07** | **68.59** |
| +TRAINSSNALT | 64.77 | 53.34 | 48.29 | 85.27 | 69.87 | 67.77 |

**Results**    When the fusion model uses the element-wise mean fusion (Table 1), TRAINSSN algorithm shows the best single source robustness against Gaussian SSN while preserving the original performance on clean data (only small decrease in the moderate BEV detection)[5]. Also a balance

of the both input sources' performance is dramatically decreased compared to the models trained without robust learning and a naive TRAINASN method.

Encouragingly, AVOD model constructed with our LEL method already achieves relatively high robustness without any robust learning strategies compared to the mean fusion layers. For all the tasks, minAP scores are dramatically increased, e.g., 61.97 vs. 47.41 for the easy 3D detection task, and the maxDiffAP scores are decreased (maxDiffAP scores for AVOD with LEL are reported in Appendix B.). Then the robustness is further improved by minimizing our MAXSSN loss. As our LEL's structure inherently handles corruption on a single source well, even the TRAINASN algorithm can successfully guide the model to be equipped with the desired robustness.

A corruption method with a different style, downsampling, is also tested with our LEL fusion method. Table 2 shows that the model trained with our TRAINSSN algorithm achieves the best robustness among the four algorithms for this complex and realistic perturbation.

**Remark 3.** A simple TRAINSSNALT achieves fairly robust models in both fusion methods against Gaussian noise, and two reasons may explain this phenomenon. First, all parameters are updated instead of fine-tuning only fusion related parts. Therefore, unlike our analyses on the linear model, the latent representation can be transformed to meet the objective function. In fact, TRAINSSNALT performs poorly when we fine-tune the model with concatenation fusion layers as shown in the supplement. Secondly, the loss function $\mathcal{L}$ inside our $\mathcal{L}_{\text{MAXSSN}}$ is usually non-convex so that it may be enough to use an indirect approach for small number of sources, $n_s = 2$.

**Remark 4.** Without applying fancier approaches which could increase computational cost, our LEL showed appealing effectiveness even with simple implementation.

## 6  Conclusion

We study two strategies to improve robustness of fusion models against single source corruption. Motivated by analyses on linear fusion models, a loss function is introduced to balance performance degradation of deep fusion models caused by corruption in different sources. We also demonstrate the importance of a fusion method's structure by proposing a simple ensemble layer achieving such robustness inherently. Our experimental results show that deep fusion models can effectively use complementary and shared information of different input sources by training with our loss and fusion layer to obtain both robustness and high accuracy. We hope our results motivate further work to improve the single source robustness of more complex fusion models with either large number of input sources or adaptive networks. Another interesting direction is to investigate the single source robustness against adversarial attacks in deep fusion models, which can be compared with our analyses in the supplementary material.

## Footnotes

[2]In practice, $Y = [X_1, X_2] \begin{bmatrix} h_1 \\ h_2 \end{bmatrix}$ has to be solved for $X_1 \in \mathbb{R}^{n \times (d_1+d_3)}, X_2 \in \mathbb{R}^{n \times (d_2+d_3)}$ and $Y \in \mathbb{R}^n$ with enough number of $n$ data samples. Then a standard least squares solution using a pseudo-inverse gives $h_1 = [\beta_1; \beta_3/2], h_2 = [\beta_2; \beta_3/2]$. This is equivalent to the solution robust against random noise added to all the sources at once, which is vulnerable to single source faults (Section 3.2).

[3]We also try *fine-tuning* only a subset of the model's parameters, $\boldsymbol{\theta}_{\text{fusion}} \subset f$, to preserve essential parts for extracting features from normal data. Although this strategy is similar to optimizing over only $g_1$ and $g_2$ in our linear fusion case, training the whole network from the beginning shows better performance in practice. See Appendix B for a detailed comparison.

[4]Our methods are implemented with TensorFlow on top of the official AVOD code. The computing machine has a Intel Xeon E5-1660v3 CPU with Nvidia Titan X Pascal GPUs. The source code is available at `https://github.com/twankim/avod_ssn`.

[5]In practice, it is difficult to identify flexible parameters related to shared information in advance, and also the design goal becomes a soft rather than a hard constraint. Therefore there is minor degradation in performance, to pay for the added robustness.

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
