[Supplementary Material]

# A Proofs and supplementary Analyses

## A.1 Proofs and analyses for Section 3.2

*Proof.* The original $\mathcal{L}_{\text{MAXSSN}}$ loss minimization problem with an additional constraint of preserving loss under clean data can be transformed to the problem stated in (3) due to the flexibility of $g_1$ and $g_2$ under the constraint $g_1 + g_2 = \beta_3$:

$$\min_{g_1, g_2} \max \left\{ \mathcal{L}\left(y, f_{\text{direct}}(x_1 + \delta_1, x_2)\right), \mathcal{L}\left(y, f_{\text{direct}}(x_1, x_2 + \delta_2)\right) \right\} \quad s.t. \ g_1 + g_2 = \beta_3$$

Under the expected squared loss with $f_{\text{direct}}$ function, the loss can be evaluated,

$$\mathcal{L}\left(y, f_{\text{direct}}(x_1 + \delta_1, x_2)\right) = \mathbb{E}\left[ \left( y - (\beta_1^T (z_1 + \epsilon_1) + g_1^T (z_3 + \epsilon_3) + \beta_2^T z_2 + g_2^T z_3) \right)^2 \right]$$

$$= \mathbb{E}\left[ \left( \beta_1^T \epsilon_1 + g_1^T \epsilon_3 \right)^2 \right] \qquad \left( \because y = \sum_{i=1}^{3} \beta_i^T z_i \right)$$

$$= \sigma^2 (||\beta_1||_2^2 + ||g_1||_2^2) \qquad (\because \text{Statistical assumption on } \epsilon_i.)$$

Hence the equivalent problem (6) is achieved.

$$\sigma^2 \min_{g_1, g_2} \max \left\{ ||\beta_1||_2^2 + ||g_1||_2^2, ||\beta_2||_2^2 + ||g_2||_2^2 \right\} \quad s.t. \ g_1 + g_2 = \beta_3 \qquad (6)$$

For simple notation, substitute variables as $g = g_1, v = \beta_3, c_1 = ||\beta_1||_2^2, c_2 = ||\beta_2||_2^2$, and solve the following convex optimization problem.

$$\min_{g} \max\{ ||g||_2^2 + c_1, ||g - v||_2^2 + c_2 \}$$

This problem can be solved by introducing a variable $\gamma$ for the upper bound of the inner maximum value:

$$\min_{g, \gamma} \gamma \quad s.t. \ c_1 + ||g||_2^2 - \gamma \le 0, \ c_2 + ||g - v||_2^2 - \gamma \le 0$$

KKT condition gives:

$$
\begin{aligned}
&\text{(Primal feasibility)} && c_1 + ||g||_2^2 - \gamma \le 0, \ c_2 + ||g - v||_2^2 - \gamma \le 0 \\
&\text{(Dual feasibility)} && \lambda_1 \ge 0, \ \lambda_2 \ge 0 \\
&\text{(Complementary slackness)} && \lambda_1(c_1 + ||g||_2^2 - \gamma) = 0, \ \lambda_2(c_2 + ||g - v||_2^2 - \gamma) = 0 \\
&\text{(Stationary)} && \lambda_1 + \lambda_2 = 1, \ g = \frac{\lambda_2}{\lambda_1 + \lambda_2} v
\end{aligned}
$$

Considering $\lambda_1 + \lambda_2 = 1$ and $\lambda_1, \lambda_2 \ge 0$, we first need to analyze the case $\lambda_1 = 0$. This gives $g = v$ and the complementary slackness condition to find $\gamma = c_2 + ||g - v||_2^2 = c_2$. $\lambda_2 = 0$ can be analyzed with similar steps. If both $\lambda_1$ and $\lambda_2$ are positive, the complementary slackness condition gives $\gamma = c_1 + ||g||_2^2 = c_2 + ||g - v||_2^2$, which ensures the balance of the original problem's maximum value $\max\{c_1 + ||g||_2^2, c_2 + ||g - v||_2^2\}$. This case gives $\gamma = \frac{c_1 + c_2}{2} + \frac{||v||_2^2}{4} + \frac{(c_2 - c_1)^2}{4||v||_2^2}$ with $g = \frac{1}{2}\left(1 + \frac{c_2 - c_1}{||v||_2^2}\right)v$. Therefore, we can have the result (4) which provides the fusion model robust against single source corruptions from random noise. $\square$

**Comparison to the model not considering MAXSSN loss** If random noise are added to $x_1$ and $x_2$ simultaneously, the objective of the problem becomes $\min_{g_1, g_2} \mathbb{E}[(y - f_{\text{direct}}(x_1 + \delta_1, x_2 + \delta_2))^2]$ instead of considering the MAXSSN loss. This is equivalent to minimizing $\sigma^2(||\beta_1||_2^2 + ||\beta_2||_2^2 + ||g_1||_2^2 + ||g_2||_2^2)$ subject to $g_1 + g_2 = \beta_3$, and the solution can be directly found as it is a simple convex problem, which is $g_1 = g_2 = \frac{\beta_3}{2}$. If we denote this model as $f'_{\text{direct}}$, then MAXSSN loss is:

$$\mathcal{L}_{\text{MAXSSN}}(f'_{\text{direct}}, \epsilon) = \mathcal{L}'_{\text{MAXSSN}} = \sigma^2 \max \left\{ ||\beta_1||_2^2 + \frac{1}{4}||\beta_3||_2^2, ||\beta_2||_2^2 + \frac{1}{4}||\beta_3||_2^2 \right\}$$

Now, let's compute the difference $\mathcal{L}'_{\text{MAXSSN}} - \mathcal{L}^*_{\text{MAXSSN}}$.

*Proof.* As both term includes $\sigma^2$, let's assume $\sigma^2 = 1$ for ease of notation. Among the three cases in (4), consider the first case $||\beta_1||_2^2 + ||\beta_3||_2^2 \leq ||\beta_2||_2^2$.

$$\mathcal{L}'_{\text{MAXSSN}} - \mathcal{L}^*_{\text{MAXSSN}} = ||\beta_2||_2^2 + \frac{1}{4}||\beta_3||_2^2 - ||\beta_2||_2^2 = \frac{1}{4}||\beta_3||_2^2 \quad (\because ||\beta_2||_2^2 \geq ||\beta_1||_2^2)$$

The second case can be shown similarly. Now assume that $\left| \frac{||\beta_2||_2^2 - ||\beta_1||_2^2}{||\beta_3||_2^2} \right| < 1$ holds, and let $||\beta_2||_2^2 \geq ||\beta_1||_2^2$ without loss of generality. Then we can show that,

$$\mathcal{L}'_{\text{MAXSSN}} - \mathcal{L}^*_{\text{MAXSSN}} = ||\beta_2||_2^2 + \frac{1}{4}||\beta_3||_2^2 - \left( \frac{||\beta_1||_2^2 + ||\beta_2||_2^2}{2} + \frac{||\beta_3||_2^2}{4} + \frac{(||\beta_2||_2^2 - ||\beta_1||_2^2)^2}{4||\beta_3||_2^2} \right)$$

$$= \frac{1}{2}(||\beta_2||_2^2 - ||\beta_1||_2^2)\left( 1 - \frac{||\beta_2||_2^2 - ||\beta_1||_2^2}{2||\beta_3||_2^2} \right)$$

$$\geq \frac{1}{4}(||\beta_2||_2^2 - ||\beta_1||_2^2) \quad \left( \because ||\beta_2||_2^2 \geq ||\beta_1||_2^2 \text{ and } \left| \frac{||\beta_2||_2^2 - ||\beta_1||_2^2}{||\beta_3||_2^2} \right| < 1 \right)$$

$\square$

Therefore we can conclude that simply optimizing under noise added to all the input sources at the same time cannot do better than minimizing MAXSSN loss with some nonnegative gap in our linear fusion model.

## A.2 Single Source Robustness against Adversarial attacks

Another important type of perturbation is an adversarial attack. Different from the previously studied random noise, perturbation to the input sources is also optimized to maximize the loss to consider the worst case. Adversarial version of the MAXSSN loss is defined as follows:

**Definition 3.** *For multiple sources $x_1, \cdots, x_{n_s}$ and a target variable $y$, denote a predefined loss function by $\mathcal{L}$. If each input source $x_i$ is maximally perturbed with some additive noise $\eta_i \in \mathcal{S}_i$ for $i \in [n_s]$, ADVMAXSSN loss for a model $f$ is defined as follows:*

$$\mathcal{L}_{\text{ADVMAXSSN}}(f, \eta) \triangleq \max_i \left\{ \max_{\eta_i \in \mathcal{S}_i} \mathcal{L}\left(y, f(x_i + \eta_i, x_{-i})\right) \right\}_{i=1}^{n_s}$$

As a simple model analysis, let's consider a binary classification problem using the logistic regression. Again, two input sources $x_1 = [z_1; z_3]$ and $x_2 = [z_2; z_3]$ have a common feature vector $z_3$ as in the linear fusion data model. A binary classifier $\text{sgn}(f(x_1, x_2))$ is trained to predict label $y \in \{-1, 1\}$, where $f(x_1, x_2) = (w_1^T z_1 + g_1^T z_3) + (w_2^T z_2 + g_2^T z_3)$ and the training loss is $\mathbb{E}_{x,y}[\ell(y \cdot f(x_1, x_2))]$ with the logistic function $\ell(x) = \log(1 + \exp(-x))$. Here, we apply one of the most popular attacks, fast gradient sign (FGS) method, which was also motivated by linear models without a fusion framework [13]. The adversarial attack $\eta_i$ per each source $x_i$ under $\ell_\infty$ norm constraint $||\eta_i||_\infty \leq \varepsilon$ can be similarly derived as follows:

$$\eta_1 = [-\varepsilon y \cdot \text{sgn}(w_1); -\varepsilon y \cdot \text{sgn}(g_1)], \quad \eta_2 = [-\varepsilon y \cdot \text{sgn}(w_2); -\varepsilon y \cdot \text{sgn}(g_2)] \quad (7)$$

As a substitute for the linear fusion data model, let's assume the true classes are generated by the hidden relationship $y = \text{sgn}\left(\sum_{i=1}^3 \beta_i^T z_i\right)$. Then the optimal fusion binary classifier becomes $\text{sgn}(f_{\text{direct}}(x_1, x_2))$. Similar to the previous section, suppose an objective is to find a model with robustness against single source adversarial attacks, while preserving the performance on clean data. Then the overall optimization problem can be reduced to the following one:

$$\min_{g_1, g_2} \max \{\mathcal{L}(y, f_{\text{direct}}(x_1 + \eta_1, x_2)), \mathcal{L}(y, f_{\text{direct}}(x_1, x_2 + \eta_2))\} \quad s.t. \ g_1 + g_2 = \beta_3 \quad (8)$$

As $\ell$ is a decreasing function, optimal $g_1$ and $g_2$ of the original problem are equivalent to the minimizer of the following one:

$$\varepsilon \min_{g_1, g_2} \max \{||w_1||_1 + ||g_1||_1, ||w_2||_1 + ||g_2||_1\} \quad s.t. \ g_1 + g_2 = \beta_3 \quad (9)$$

By solving this convex optimization problem, we can achieve solution $\mathcal{L}^*_{\text{ADVMAXSSN}}$ and optimizers $g_1^*, g_2^*$. Also, we can find $\mathcal{L}'_{\text{ADVMAXSSN}}$, a $\mathcal{L}_{\text{ADVMAXSSN}}$ value evaluated using the optimal model for

minimizing the adversarial attacks added to all the sources at once. Interestingly, we can show that $\mathcal{L}'_{\text{ADVMAXSSN}} \geq \mathcal{L}^*_{\text{ADVMAXSSN}}$ if $\frac{||\beta_2||_1 - ||\beta_1||_1}{||\beta_3||_1} > 1$, but $\mathcal{L}'_{\text{ADVMAXSSN}} = \mathcal{L}^*_{\text{ADVMAXSSN}}$ otherwise. In other words, if inherent influence of $z_1$ and $z_2$ are well balanced compared to the common feature $z_3$ in the sense of $\ell_1$ norm, adversarial attacks only applied to a single source can be equivalently defended by just using a traditional adversarial training strategy to learn a model robust against attacks added to all the sources at once.

*Proof.* The original minimizing $\mathcal{L}_{\text{ADVMAXSSN}}$ loss minimization problem with an additional constraint of preserving loss under clean data can be transformed to the problem stated in (8) due to the flexibility of $g_1$ and $g_2$:

$$\min_{g_1, g_2} \max \left\{ \mathcal{L}\left(y, f_{\text{direct}}(x_1 + \eta_1, x_2)\right), \mathcal{L}\left(y, f_{\text{direct}}(x_1, x_2 + \eta_2)\right) \right\} \quad s.t. \ g_1 + g_2 = \beta_3$$

As $\eta_i$'s are assumed to be made with FGS method, adversarial attacks under $\ell_\infty$ norm constraints are as follows:

$$\eta_1 = [-\varepsilon y \cdot \text{sgn}(w_1); -\varepsilon y \cdot \text{sgn}(g_1)], \ \eta_2 = [-\varepsilon y \cdot \text{sgn}(w_2); -\varepsilon y \cdot \text{sgn}(g_2)]$$

Therefore, minimizing $\mathcal{L}_{\text{ADVMAXSSN}}(f_{\text{direct}}, \eta)$ over $g_1, g_2$ becomes:

$$\min_{g_1, g_2} \max \{ \mathbb{E}\left[\ell\left(y \cdot f_{\text{direct}}(x_1, x_2) - \varepsilon(||w_1||_1 + ||g_1||_1)\right)\right],$$
$$\mathbb{E}\left[\ell\left(y \cdot f_{\text{direct}}(x_1, x_2) - \varepsilon(||w_2||_1 + ||g_2||_1)\right)\right]\} \quad s.t. \ g_1 + g_2 = \beta_3$$

We can solve the following problem to find minimizers $g_1^*$ and $g_2^*$.

$$\min_{g_1, g_2} \max \{||w_1||_1 + ||g_1||_1, ||w_2||_1 + ||g_2||_1\} \quad s.t. \ g_1 + g_2 = \beta_3$$

Similar to the random noise case, substitute variables as $g = g_1, v = \beta_3, c_1 = ||\beta_1||_1, c_2 = ||\beta_2||_2$, and solve the following convex optimization problem:

$$\min_g \max \{||g||_1 + c_1, ||g - v||_1 + c_2\}$$

which can be solved by introducing $\gamma$,

$$\min_{g, \gamma} \gamma \quad s.t. \ c_1 + ||g||_1 - \gamma \leq 0, \ c_2 + ||g - v||_1 - \gamma \leq 0$$

KKT condition gives:

| | |
|---|---|
| (Primal feasibility) | $c_1 + ||g||_1 - \gamma \leq 0, \ c_2 + ||g - v||_1 - \gamma \leq 0$ |
| (Dual feasibility) | $\lambda_1 \geq 0, \ \lambda_2 \geq 0$ |
| (Complementary slackness) | $\lambda_1(c_1 + ||g||_1 - \gamma) = 0, \ \lambda_2(c_2 + ||g - v||_1 - \gamma) = 0$ |
| (Stationary) | $\lambda_1 + \lambda_2 = 1, \ 0 \in \lambda_1 \partial ||g||_1 + \lambda_2 \partial ||g - v||_1$ |

If $\lambda_1 = 0$ or $\lambda_2 = 0$, these cases handle when the inherent imbalance of three components $z_1, z_2$ and $z_3$. Consider $\lambda_2 = 0$, which gives $||g||_1 + c_1 - \gamma = 0$ from the complementary slackness condition. And the stationary condition becomes $0 \in \partial ||g||_1$. As a subgradient of $||g||_1$ can be zero if and only if $g(i) = 0$ for any $i^{th}$ component, the solution is $g = 0$ with $\gamma = c_1$ and the necessary condition is $||v||_1 + c_2 \leq c_1$. Similar solution can be found for $\lambda_1 = 0$ case as $g = v, \gamma = c_2$ if $||v||_1 + c_1 \leq c_2$. Therefore, we can have $\gamma^* = \min \max \{||w_1||_1 + ||g_1||_1, ||w_2||_1 + ||\beta_3 - g_1||_1\}$ and corresponding parameters as:

$$(\gamma^*, g_1^*, g_2^*) = \begin{cases} (||\beta_2||_1, \beta_3, 0) & \text{if } ||\beta_1||_1 + ||\beta_3||_1 \leq ||\beta_2||_1 \\ (||\beta_1||_1, 0, \beta_3) & \text{if } ||\beta_2||_1 + ||\beta_3||_1 \leq ||\beta_1||_1 \end{cases}$$

Now let's consider $\lambda_1 \neq 0, \lambda_2 \neq 0$. Denote $q \in \lambda_1 \partial ||g||_1 + \lambda_2 \partial ||g - v||_1$ as the element of subdifferential of the Lagrangian. We need to find cases for $q(i) = 0$ to hold.

(i) If $v(i) = 0$, then $\text{sgn}(g(i)) = \text{sgn}(g(i) - v(i))$ holds. Therefore, if $g(i) \neq = 0$, a subgradient becomes $q(i) = \lambda_1 \text{sgn}(g(i)) + \lambda_2 \text{sgn}(g(i)) = \text{sgn}(g(i))$ which cannot be zero. $\Rightarrow g(i) = v(i) = 0$.

(ii) If $v(i) \neq 0$, we need to consider three different sub cases. First, if $g(i) \neq 0$ and $g(i) \neq v(i)$, then $q(i) = \lambda_1(\mathrm{sgn}(g(i)) - \mathrm{sgn}(g(i) - v(i))) + \mathrm{sgn}(g(i) - v(i))$. For $q(i) = 0$ to hold, $\mathrm{sgn}(g(i)) = -\mathrm{sgn}(g(i) - v(i))$ must be true with $\lambda_1 = \frac{1}{2}$. This gives a solution $g(i) = \alpha_i v(i)$ with $\forall \alpha_i \in (0, 1)$.

Secondly, if $g(i) = 0$ but $g(i) \neq v(i)$, then the subgradient is $q(i) = \lambda_1 \alpha_i + (1 - \lambda_1)\mathrm{sgn}(-v(i))$ for any $\alpha_i \in [-1, 1]$. Therefore, if $\alpha_i = \frac{1-\lambda_1}{\lambda_1}\mathrm{sgn}(v(i))$ with some $\lambda_1 \in [\frac{1}{2}, 1)$, the stationary condition holds.

Finally, if $g(i) \neq 0$ and $g(i) = v(i)$, then $q(i) = \lambda_1 \mathrm{sgn}(g(i)) + (1 - \lambda_1)\alpha_i$ for any $\alpha_i \in [-1, 1]$. Therefore, if $\alpha_i = \frac{\lambda_1}{1-\lambda_1}\mathrm{sgn}(v(i))$ with $\lambda_1 \in (0, \frac{1}{2}]$, $q(i) = 0$ holds for the stationary condition.

All the above cases in (i) and (ii) can be restated as a combined solution $g(i) = \alpha_i v(i), \forall \alpha_i \in [0, 1]$. It is easy to show that $|g(i)| + |g(i) - v(i)| = |v(i)|$ holds for any $i$. Also, $\lambda_1 \neq 0, \lambda_2 \neq 0$ with the complementary slackness condition gives a new constraint $\gamma = ||g||_1 + c_1 = ||g - v||_1 + c_2$. Hence, we can calculate $\gamma$ by averaging the two equivalent values:

$$\gamma = \frac{1}{2}(c_1 + c_2 + ||g||_1 + ||g - v||_1) = \frac{1}{2}(c_1 + c_2 + ||v||_1)$$

Therefore, $(\gamma^*, g_1^*, g_2^*) = \left(\frac{1}{2}(||\beta_1||_1 + ||\beta_2||_1 + ||\beta_3||_1), \alpha \odot \beta_3, \beta_3 - \alpha \odot \beta_3\right)$, where $\odot$ is an element-wise product and each element of $\alpha$ can have any value in $[0, 1]$, i.e. $\alpha(i) \in [0, 1]$.

Now, let's consider a model robust against adversarial attacks added to both sources $x_1$ and $x_2$ at the same time. This becomes a problem of minimizing $||\beta_1||_1 + ||\beta_2||_1 + ||g_1||_1 + ||\beta_3 - g_1||_1$. And the optimal solution can be achieved by $(g_1', g_2') = (\alpha \odot \beta_3, \beta_3 - \alpha \odot \beta_3)$ for any alpha satisfying $\alpha(i) \in [0, 1]$. Therefore, we can conclude that our $\mathcal{L}_{\text{ADVMAXSSN}}$ loss is necessary to give a binary classifier more robust against single source adversarial attacks, i.e. $\mathcal{L}_{\text{ADVMAXSSN}}^* \leq \mathcal{L}_{\text{ADVMAXSSN}}'$, if $\frac{||\beta_2||_1 - ||\beta_1||_1}{||\beta_3||_1} > 1$ holds. Surprisingly, if $\frac{||\beta_2||_1 - ||\beta_1||_1}{||\beta_3||_1} \leq 1$ holds to have balanced influence from inherent components from the different source of inputs, $\mathcal{L}_{\text{ADVMAXSSN}}^* = \mathcal{L}_{\text{ADVMAXSSN}}'$. In other words, if different input sources contributes to the target variable with certain balance, a traditional way of generating adversarial samples by considering all the sources at once can train a model robust against single source attacks as well. $\square$

## B   Additional Experimental Results

**Evaluation on ASN data**   Although our main focus is corruption on a single source, it is possible for a model to encounter a case where all the sources are corrupted. If the level of corruption is severe, then extracting any meaningful information from the input sources is impossible, e.g. occlusion on every sensors. However, we hope our model to be robust against reasonably corrupted input sources even if our training objective leans toward the single source robustness. Therefore, we also report the model's performance against data corrupted with ASN. In most cases, the AVOD learned with TRAINASN method achieves the best robustness against ASN, which is designed to do so. However, a model using element-wise mean fusion layers trained with TRAINASN shows lower robustness scores compared to the SSN oriented approaches. We believe that this phenomenon is caused by corrupted feature extraction combined with the structural limitation of the mean fusion layer.

**Fine-tuning**   We also consider another algorithmic framework using *fine-tuing*. The algorithm starts with a normal training on clean data for $m_{\text{clean}}$ iterations, which may include some general data augmentation methods like random cropping, and flipping. Then $m_{\text{tune}}$ steps of fine-tuning is run to update only a subset of the model's parameters, $\boldsymbol{\theta}_{\text{fusion}} \subset f$, so that any essential parts for extracting features from normal data are not affected. Convolutional layers extracting features from different sources before the fusion stages are fixed, and other layers for fusing the features and making predictions are updated in the fine-tuning stage. The experimental results using this method are provided in Table 4 and 6 for the Gaussian noise case. Overall performance of the fusion model trained from the scratch is better than using fine-tuning. This shows the importance of feature extraction parts in deep learning models.

**Concatenation**   Our analyses in Section 3 assume to use a linear fusion model with a simple concatenation strategy. Therefore, we first train the AVOD model with concatenation fusion layers on clean data and fine-tune with different training strategies. Interestingly, a simple data augmentation

Table 3: Car detection (3D/BEV) performance of AVOD with *element-wise mean* fusion layers against *Gaussian* SSN and ASN on the KITTI validation set.

| (Data) Train Algo. | Easy | Moderate | Hard | Easy | Moderate | Hard |
|---|---|---|---|---|---|---|
| (Clean Data) | $AP_{3D}(\%)$ | | | $AP_{BEV}(\%)$ | | |
| AVOD [25] | 76.41 | **72.74** | **66.86** | 89.33 | 86.49 | **79.44** |
| +TRAINASN | 75.96 | 66.68 | 65.97 | 88.63 | 79.45 | 78.79 |
| +TRAINSSN | 76.28 | 67.10 | 66.51 | 88.86 | 79.60 | 79.11 |
| +TRAINSSNALT | **77.46** | 67.61 | 66.06 | **89.68** | **86.71** | 79.41 |
| (Gaussian ASN) | $AP_{3D}(\%)$ | | | $AP_{BEV}(\%)$ | | |
| AVOD [25] | 28.08±0.91 | 26.35±2.18 | 21.81±0.63 | 42.01±0.23 | 33.68±0.17 | 33.60±0.13 |
| +TRAINASN | 61.26±0.45 | 47.71±0.24 | 45.60±0.19 | 87.40±0.07 | 72.07±2.89 | 70.13±0.05 |
| +TRAINSSN | 69.33±0.43 | 55.41±0.21 | **52.90±2.12** | **88.39±0.13** | **78.37±0.10** | 70.75±0.05 |
| +TRAINSSNALT | **71.63±0.04** | **56.24±0.16** | 49.14±0.10 | 87.95±0.08 | 77.88±0.17 | 69.96±0.08 |
| (Gaussian SSN) | $\min AP_{3D}(\%)$ | | | $\min AP_{BEV}(\%)$ | | |
| AVOD [25] | 47.41±0.28 | 41.84±0.17 | 36.47±0.16 | 65.63±0.28 | 58.02±0.23 | 50.43±0.14 |
| +TRAINASN | 61.53±0.57 | 52.72±0.08 | 47.25±0.13 | 87.71±0.14 | 78.37±0.06 | 77.85±0.08 |
| +TRAINSSN | 71.65±0.31 | **62.14±0.08** | **56.78±0.12** | 88.21±0.08 | 78.90±0.09 | **77.92±0.11** |
| +TRAINSSNALT | **71.66±0.48** | 57.61±0.12 | 55.90±0.11 | **89.42±0.04** | **79.56±0.06** | 77.92±0.05 |
| (Gaussian SSN) | $\max \mathrm{Diff} AP_{3D}(\%)$ | | | $\max \mathrm{Diff} AP_{BEV}(\%)$ | | |
| AVOD [25] | 26.70±0.52 | 22.42±0.29 | 20.92±0.25 | 22.27±0.41 | 20.76±0.33 | 20.09±0.20 |
| +TRAINASN | 14.48±0.82 | 12.72±0.33 | 11.18±0.27 | 0.88±0.22 | 0.48±0.13 | 0.28±0.12 |
| +TRAINSSN | **3.71±0.46** | **3.42±0.25** | 7.50±0.25 | 0.36±0.17 | **0.04±0.15** | 0.71±0.17 |
| +TRAINSSNALT | 5.55±0.81 | 8.73±0.32 | **2.91±0.22** | **0.09±0.14** | 0.13±0.11 | **0.18±0.11** |

Table 4: Car detection (3D/BEV) performance of AVOD with *element-wise mean* fusion layers (trained with fine-tuning) against *Gaussian* SSN and ASN on the KITTI validation set.

| (Data) Train Algo. | Easy | Moderate | Hard | Easy | Moderate | Hard |
|---|---|---|---|---|---|---|
| (Clean Data) | $AP_{3D}(\%)$ | | | $AP_{BEV}(\%)$ | | |
| AVOD [25] | **76.41** | **72.74** | **66.86** | **89.33** | **86.49** | **79.44** |
| +TRAINASN | 62.55 | 55.81 | 55.34 | 79.08 | 69.90 | 69.83 |
| +TRAINSSN | 73.50 | 65.66 | 64.74 | 88.27 | 85.65 | 78.98 |
| +TRAINSSNALT | 75.76 | 71.99 | 66.31 | 88.76 | 85.73 | 79.14 |
| (Gaussian ASN) | $AP_{3D}(\%)$ | | | $AP_{BEV}(\%)$ | | |
| AVOD [25] | 28.08±0.91 | 26.35±2.18 | 21.81±0.63 | 42.01±0.23 | 33.68±0.17 | 33.60±0.13 |
| +TRAINASN | **68.58±1.93** | **54.76±0.30** | **48.00±0.29** | **83.15±3.01** | **76.10±0.069** | **68.49±0.08** |
| +TRAINSSN | 60.73±0.32 | 45.52±0.19 | 44.42±0.11 | 78.24±0.10 | 68.41±0.10 | 60.45±0.07 |
| +TRAINSSNALT | 53.25±0.27 | 44.96±0.14 | 38.64±0.10 | 68.69±0.18 | 59.41±0.14 | 51.37±0.07 |
| (Gaussian SSN) | $\min AP_{3D}(\%)$ | | | $\min AP_{BEV}(\%)$ | | |
| AVOD [25] | 47.41±0.28 | 41.84±0.17 | 36.47±0.16 | 65.63±0.28 | 58.02±0.23 | 50.43±0.14 |
| +TRAINASN | 52.72±0.34 | 45.66±0.24 | 39.29±0.22 | 69.33±0.21 | 60.19±0.15 | 59.66±0.15 |
| +TRAINSSN | 62.46±0.48 | 53.85±0.22 | 47.62±0.14 | 77.77±0.16 | 68.71±0.09 | 67.89±0.09 |
| +TRAINSSNALT | **70.09±0.46** | **56.20±0.21** | **54.46±0.13** | **84.46±2.66** | **76.32±0.06** | **68.74±0.08** |

strategy TRAINSSNALT does not work well in this case, and TRAINASN algorithm learns the best robust model. Unlike our simple linear model deep learning jointly learns both feature representation and weights for the fusion layers. Also, concatenated convolutional features have large number of channels which are mixed without sparse constraints. Therefore, this may lead to a model with too complex joint feature representation which needs stronger guideline in optimization steps.

**Results on downsampling corruption** Downsampling the LIDAR sensor is important as it is not clear whether a model trained with a high-resolution sensor will still work with a low-resolution one. In fact, reducing the number of lasers of a LIDAR is directly related to its price, which an important practical issue in deploying an actual autonomous vehicle. As the rotating LIDAR sensor used in the KITTI dataset outputs point clouds with a horizontal structure, an RGB image's horizontal lines are also set to black to match the information loss ratio 1/4. Table 8 fully reports the performance of AVOD using our LEL when downsampling is considered as a corruption method.

Table 5: Car detection (3D/BEV) performance of AVOD with *latent ensemble layers (LEL)* against *Gaussian* SSN and ASN on the KITTI validation set.

| (Data) Train Algo. | Easy | Moderate | Hard | Easy | Moderate | Hard |
|---|---|---|---|---|---|---|
| (Clean Data) | $AP_{3D}(\%)$ | | | $AP_{BEV}(\%)$ | | |
| AVOD [25] | **77.79** | **67.69** | **66.31** | **88.90** | **85.64** | **78.86** |
| +TRAINASN | 75.00 | 64.75 | 58.28 | 88.30 | 78.60 | 77.23 |
| +TRAINSSN | 74.25 | 65.00 | 63.83 | 87.88 | 78.84 | 77.66 |
| +TRAINSSNALT | 76.04 | 66.42 | 64.41 | 88.80 | 79.53 | 78.53 |
| (Gaussian ASN) | $AP_{3D}(\%)$ | | | $AP_{BEV}(\%)$ | | |
| AVOD [25] | 46.79±0.37 | 41.46±0.27 | 36.31±0.20 | 77.40±0.34 | 67.46±0.11 | 59.53±0.11 |
| +TRAINASN | **74.24±0.29** | **63.47±0.18** | **57.25±0.19** | 87.72±0.12 | **77.89±0.09** | **70.36±0.05** |
| +TRAINSSN | 67.69±0.28 | 55.74±0.30 | 53.16±0.32 | **87.73±0.16** | 77.80±0.15 | 70.00±0.10 |
| +TRAINSSNALT | 63.72±0.40 | 53.15±0.29 | 48.17±0.22 | 85.36±0.08 | 75.60±0.08 | 69.17±0.03 |
| (Gaussian SSN) | min $AP_{3D}(\%)$ | | | min $AP_{BEV}(\%)$ | | |
| AVOD [25] | 61.97±0.55 | 53.95±0.42 | 47.24±0.27 | 79.44±0.09 | 72.46±3.14 | 68.25±0.06 |
| +TRAINASN | **74.24±0.38** | 58.25±0.16 | **56.13±0.10** | 88.10±0.26 | **78.19±0.13** | 70.42±0.07 |
| +TRAINSSN | 68.16±0.88 | **60.39±0.38** | 56.04±0.28 | **88.12±0.16** | 78.17±0.06 | 70.21±0.05 |
| +TRAINSSNALT | 68.63±0.40 | 55.48±0.16 | 54.42±0.17 | 86.51±0.46 | 76.85±0.11 | **71.95±2.72** |
| (Gaussian SSN) | max Diff$AP_{3D}(\%)$ | | | max Diff$AP_{BEV}(\%)$ | | |
| AVOD [25] | 3.75±2.05 | 0.98±0.55 | 5.95±0.40 | 7.28±0.37 | 4.46±3.25 | **1.25±0.13** |
| +TRAINASN | **1.54±0.40** | **0.85±0.24** | **0.83±0.25** | 0.92±0.17 | 1.09±0.14 | 7.44±0.08 |
| +TRAINSSN | 4.61±1.16 | 2.51±0.50 | 0.74±0.46 | **0.16±0.32** | **0.72±0.14** | 7.10±0.14 |
| +TRAINSSNALT | 4.65±1.04 | 7.88±0.46 | 2.90±0.45 | 1.12±0.71 | 1.83±0.17 | 3.42±2.84 |

Table 6: Car detection (3D/BEV) performance of AVOD with *latent ensemble layers (LEL)* (trained with fine-tuning) against *Gaussian* SSN and ASN on the KITTI validation set.

| (Data) Train Algo. | Easy | Moderate | Hard | Easy | Moderate | Hard |
|---|---|---|---|---|---|---|
| (Clean Data) | $AP_{3D}(\%)$ | | | $AP_{BEV}(\%)$ | | |
| AVOD [25] | **77.79** | **67.69** | **66.31** | **88.90** | **85.64** | 78.86 |
| +TRAINASN | 74.65 | 65.40 | 63.40 | 88.18 | 79.21 | 78.42 |
| +TRAINSSN | 76.95 | 67.22 | 65.66 | 88.77 | 79.74 | **78.96** |
| +TRAINSSNALT | 76.81 | 67.46 | 66.12 | 88.47 | 79.62 | 78.86 |
| (Gaussian ASN) | $AP_{3D}(\%)$ | | | $AP_{BEV}(\%)$ | | |
| AVOD [25] | 46.79±0.37 | 41.46±0.27 | 36.31±0.20 | 77.40±0.34 | 67.46±0.11 | 59.53±0.11 |
| +TRAINASN | **63.73±0.24** | **53.16±0.16** | **47.79±0.17** | **80.18±0.07** | **76.26±0.03** | **69.12±0.04** |
| +TRAINSSN | 60.80±0.48 | 47.73±0.13 | 45.67±0.15 | 79.82±0.22 | 69.66±0.10 | 68.38±0.10 |
| +TRAINSSNALT | 52.25±1.47 | 43.77±0.62 | 37.91±0.48 | 77.51±0.12 | 67.32±0.09 | 59.65±0.10 |
| (Gaussian SSN) | min $AP_{3D}(\%)$ | | | min $AP_{BEV}(\%)$ | | |
| AVOD [25] | 61.97±0.55 | 53.95±0.42 | 47.24±0.27 | 79.44±0.09 | 72.46±3.14 | 68.25±0.06 |
| +TRAINASN | **68.08±0.44** | **57.28±0.18** | **55.27±0.20** | 86.45±0.08 | 77.19±0.08 | 69.57±0.08 |
| +TRAINSSN | 67.98±1.31 | 55.61±0.23 | 53.76±0.20 | **86.87±0.12** | **77.56±0.05** | **69.81±0.08** |
| +TRAINSSNALT | 62.76±0.41 | 52.14±0.26 | 46.55±0.13 | 85.34±2.36 | 75.72±0.04 | 68.60±0.02 |

Table 7: Car detection (3D/BEV) performance of AVOD with *concatenation* fusion layers (trained with fine-tuning) against *Gaussian* SSN and ASN on the KITTI validation set.

| (Data) Train Algo. | Easy | Moderate | Hard | Easy | Moderate | Hard |
|---|---|---|---|---|---|---|
| (Clean Data) | $AP_{3D}(\%)$ | | | $AP_{BEV}(\%)$ | | |
| AVOD [25] | **78.40** | **74.88** | **67.78** | **89.74** | **87.76** | **79.83** |
| +TRAINASN | 72.89 | 63.47 | 62.22 | 88.44 | 84.97 | 78.88 |
| +TRAINSSN | 76.15 | 66.79 | 65.78 | 89.02 | 86.06 | 79.29 |
| +TRAINSSNALT | 76.46 | 72.98 | 66.94 | 89.07 | 86.39 | 79.34 |
| (Gaussian ASN) | $AP_{3D}(\%)$ | | | $AP_{BEV}(\%)$ | | |
| AVOD [25] | 16.50±2.27 | 15.12±0.06 | 15.06±0.08 | 25.81±0.23 | 25.38±0.18 | 17.45±0.08 |
| +TRAINASN | **69.21±0.24** | **54.85±0.08** | **53.30±0.08** | **86.07±0.11** | **76.42±0.04** | **69.54±0.02** |
| +TRAINSSN | 62.05±0.36 | 50.35±2.58 | 46.04±0.25 | 79.21±0.08 | 69.31±0.10 | 61.21±0.06 |
| +TRAINSSNALT | 33.86±2.85 | 27.99±0.64 | 22.59±0.60 | 42.65±0.18 | 41.77±0.18 | 34.13±0.12 |
| (Gaussian SSN) | min $AP_{3D}(\%)$ | | | min $AP_{BEV}(\%)$ | | |
| AVOD [25] | 31.23±0.31 | 30.27±0.13 | 30.49±0.18 | 43.04±0.16 | 42.81±0.10 | 42.96±0.08 |
| +TRAINASN | **68.21±0.37** | 54.50±0.26 | 47.91±0.21 | **86.66±0.11** | **76.95±0.11** | **69.70±0.08** |
| +TRAINSSN | 64.39±0.23 | **55.12±0.21** | **48.38±0.14** | 79.71±0.07 | 70.05±0.07 | 69.32±0.10 |
| +TRAINSSNALT | 44.25±0.49 | 37.23±0.44 | 37.58±0.34 | 59.06±0.12 | 51.19±0.08 | 51.28±0.06 |

Table 8: Car detection (3D/BEV) performance of AVOD with *latent ensemble layers (LEL)* against *downsampling* SSN and ASN on the KITTI validation set.

| (Data) Train Algo. | Easy | Moderate | Hard | Easy | Moderate | Hard |
|---|---|---|---|---|---|---|
| (Clean Data) | $AP_{3D}(\%)$ | | | $AP_{BEV}(\%)$ | | |
| AVOD [25] | **77.79** | **67.69** | **66.31** | 88.90 | **85.64** | **78.86** |
| +TRAINASN | 71.74 | 61.78 | 60.26 | 87.29 | 77.08 | 75.89 |
| +TRAINSSN | 75.54 | 66.26 | 63.72 | 88.07 | 79.18 | 78.03 |
| +TRAINSSNALT | 76.22 | 66.05 | 63.87 | **89.00** | 79.65 | 78.03 |
| (Downsample ASN) | $AP_{3D}(\%)$ | | | $AP_{BEV}(\%)$ | | |
| AVOD [25] | 36.13 | 27.39 | 26.39 | 77.60 | 59.84 | 51.82 |
| +TRAINASN | **71.30** | **56.04** | **49.08** | 85.66 | **70.17** | **68.55** |
| +TRAINSSN | 64.88 | 48.92 | 47.06 | **86.21** | 69.26 | 61.48 |
| +TRAINSSNALT | 48.98 | 36.30 | 31.06 | 75.00 | 51.35 | 49.60 |
| (Downsample SSN) | min $AP_{3D}(\%)$ | | | min $AP_{BEV}(\%)$ | | |
| AVOD [25] | 61.70 | 51.66 | 46.17 | 86.08 | 69.99 | 61.55 |
| +TRAINASN | 65.74 | 53.49 | 51.35 | 82.27 | 67.88 | 65.79 |
| +TRAINSSN | **73.33** | **57.85** | **54.91** | **86.61** | **76.07** | **68.59** |
| +TRAINSSNALT | 64.77 | 53.34 | 48.29 | 85.27 | 69.87 | 67.77 |
| (Downsample SSN) | max Diff$AP_{3D}(\%)$ | | | max Diff$AP_{BEV}(\%)$ | | |
| AVOD [25] | 11.71 | 5.88 | 3.59 | 1.96 | 7.60 | 8.65 |
| +TRAINASN | 10.00 | 11.34 | 11.76 | 6.53 | 11.23 | 12.40 |
| +TRAINSSN | **0.94** | 5.71 | 3.11 | 1.74 | 2.36 | 9.00 |
| +TRAINSSNALT | 6.98 | **3.63** | **1.34** | **1.67** | **0.12** | **0.81** |