[Reviews · NeurIPS 2019]

Reviewer 1



Originality: To the best of my knowledge the proposed approach is original. The new loss MaxSSN and the new training strategy does seem to significantly outperform the state of the art AVOD in all experiments. Same with the latent ensemble layer, Clarity: The paper is clearly written and easy to follow. Quality: Although the propose method is simple, it seems effective. Technically the methods seems sounds and works. However, it may actually work only for the case demonstrated where there are two sources of information being fused. It remains to see whether it will work in a different setting, where sources of information have some other dependencies, or may even be correlated. Also, what happens when there are more than two sources of information. Significance: The proposed method is significant enough to merit publication.

Reviewer 2



Originality. Pros: To the best of my knowledge, the proposed method is new. Quality. Pros: - Detailed proofs and analysis of the proposed method are provided. - Ablation study is provided for a good understanding of the work. - Experimental results are good. Cons: - The paper claims that the method preserves the original performance on clean data. However, there are evident performance drop on clean data after applying the TrainSSN algorithm. For example, the 3D AP and BEV AP decrease by ~5% and ~7% for moderate clean data in Table 1. - For demonstration of the generalization ability of the method, it would be better if the paper could validate on more tasks/models. Currently only one task (3D detection) and one model (AVOD) is validated. Clarity. Pros: - In overall, the paper is well written and organized. Cons: - The hyper-parameter (t) in the sparse constrain of the latent ensemble layer is missing. Significance. Pros: The robust learning problem in feature fusion model that the paper addresses is an important topic and critical in practical applications like autonomous driving. I think the proposed method and results should be useful for the community.

Reviewer 3



Summary This paper discusses the importance and the method for deep fusion model with single-source noise with experiments on 3D/BEV object detection. It first proposes a novel loss called MAXSSN, as a loss used in the whole paper for single-source robustness. It then shows the limitation of standard robust fusion model -- if we do not consider every single loss separately -- adding all of them to the input at once, we would get a worse model. Two algorithms are proposed for minimizing the MAXSSN loss. The basic idea is to alternatively train on clean data and data with noise. The authors then provide a feature fusion method to ensemble feature maps from different input sources. It is assumed that all feature maps have the same width and height but (possibly) different number of channels. The basic idea of the feature fusion method is to concatenate all the feature maps and then apply a convolutional layer with 1x1 kernel. The author finally shows the experiment on KITTI dataset, doing 3D/BEV object detection. The results indicate when the data contain single-source noise, the proposed method has better performance. Strengths -The problem of robustness is important, and single-source robustness is novel. -The paper is clearly written, with almost all of the variables/notations clearly defined. -This paper provides plenty of experiments (in the main text and supplementary), showing good performance of the proposed method. Weaknesses -Analysis for algorithm 1 and 2 are proposed without much theoretical analysis. They are “suddenly” proposed and then followed by the experiments, without illustration about why the algorithms are designed in this way, or their theoretical analysis. For example, why do we have to alternate between clean data and data with noise? Algorithm 2 ignores the maximum loss, so why does it work? -The proposed latent ensemble layer is not a very novel way to fuse features. It concatenates all the feature maps and then applies a 1x1 convolutional layer, which is a reasonable way of fusion but without much novelty. -The problem of robustness is definitely important, but from my understanding, usually, noise does not come from a single source but at least a few of input sources. Can the proposed method be easily generalized? If not, we may need more illustration about the importance of single-source robustness, about why we should focus on single-source noise. Minor issues -Equation 4: some of the 2-norm subscripts are missing -Line 8 of algorithm 1: on the right-hand side you may use the notation you introduced in line 6 Comments after the rebuttal ************************************ Thank all the authors for the rebuttal. I am satisfied with the authors' explanation about the importance of single-source robustness. But judging from the rebuttal, the authors do not have a theoretical proof about TrainSSN and TrainSSNAlt (e.g., convergence, an upper bound of loss). In summary, the paper is the first one to formalize single-source robustness and provides two baseline algorithms without proof, which is acceptable. I believe it is marginally above the acceptance threshold, so I raised my score to 6.

[Author Response · NeurIPS 2019]

We thank all reviewers for their constructive feedback and detailed comments. Our responses are provided below.

**To Reviewer 1**   Thanks for your supportive and helpful comments. We also believe that applying our methods on
various fusion settings is an interesting direction for future research (Conclusion).

**Q:** Other dependencies of input sources, e.g., correlated information.
**A:** Using multiple camera inputs from various viewpoints can simulate an extremely correlated case. In fact, our
experiments deal with correlated information. Object proposals or information about potential surrounding objects can
be extracted from both RGB and LIDAR data, and *shared information* is introduced to consider such correlation.

**Q:** How our methods scale with more than two input sources.
**A:** Considering its definition, our MAXSSN loss will still try to balance single source robustness for multiple sensory
inputs. On computational complexity, our TRAINSSN requires $n_s + 1$ forward passes and 1 back propagation. One can
further improve on speed by approximating the loss, e.g., mixing TRAINSSN and TRAINSSNALT.

**To Reviewer 2**   We would like to appreciate your valuable and encouraging comments.

**Q:** Degraded performance of our methods on clean data.
**A:** Retaining performance on clean data, is an idealistic design goal. In section 3, we solve problem (3) by optimizing
over flexible parameters $g_1$ and $g_2$. If the parts of input sources contributing to $z_3$ are known, then indeed we can
achieve this goal. In practice however, it is difficult to know which parts of an input source (or latent representation) are
related to shared information and which parameters are flexible, and also the design goal becomes a soft rather than a
hard constraint. Therefore there is minor degradation in performance, to pay for the added robustness. We will add this
discussion in our paper.

**Q:** More tasks/models to demonstrate the generalization ability of the work
**A:** AVOD was selected because it is the leading deep fusion model for 3D object detection, and this model has already
been favorably compared with prior approaches, so we did not want to take up space with additional comparisons. 3D
detection is both an important problem in self-driving cars and one where multiple sensors can contribute fruitfully by
providing both complementary and shared information. In contrast, models for 2D object detection heavily rely on
RGB data, which typically dominates other modalities. Trying our approaches on other tasks like audio-visual speech
recognition is worthwhile, but our paper is already at the page limit covering more basic aspects, e.g., comparison of
fusion methods, training algorithms, and corruption methods.
We set the hyper-parameter for $\ell_1$ constraint as 0.01, and is available in our source code (omitted for anonymity).

**To Reviewer 3**   Thanks for your constructive comments and valuable time in understanding our work.

**Q:** More analyses of Algorithm 1 and 2 and explain why TRAINSSNALT works?
**A:** We can certainly add more explanation. Essentially the approaches are motivated by findings from the simple linear
fusion model, as well as data augmentation. Alternating between clean data and corrupted data aims at increased
robustness without much degradation of performance on clean data (strategy is supported by experimental results).
We also tried *fine-tuning* only fusion related layers to preserve essential parts for normal data, but Algorithm 1 and
2 are selected for better performance. We discuss the reasons for superior performance of Algorithm 2 in Remark 3.
In particular, an element-wise mean operation restricts the features to be optimal for averaging, and hence updating
without the maximum loss helps balance the performance. Also note that when AVOD with *concatenation* fusion is
trained with TRAINSSNALT, it works a lot worse than TRAINSSN.

**Q:** The proposed latent ensemble layer is not a very novel approach
**A:** When we devised our LEL, we first set the three objectives (1st sentence of the 2nd paragraph of Section 4.2): (i) error
reduction of ensemble methods, (ii) admitting source-specific features to survive, and (iii) allowing different channel
depths. Then we wrote the equation in Figure 1 and noted that it can be implemented by using $1 \times 1$ convolution.
Without applying fancier approaches which could increase computational cost, our LEL showed appealing effectiveness
even with simple implementation.

**Q:** Importance of single source robustness and the generalization ability of the model.
**A:** Currently there are no formulations even for single source, and our research is the first step. This framework can
be extended to robustness against corruption in a subset of input sources, e.g., generalizing to robustness given $k$ of
$n_s$ corrupted sources. Self-driving cars using an RGB camera and ranging sensors like LIDAR and radar are exposed
to single source corruption. For example, Uber's fatal self-driving crash occurred in nighttime and a camera couldn't
detect the victim because of darkness, while LIDAR and radar was not affected. In adversarial settings too, single
source attacks seem more feasible. Situations with multiple failures (outside of combat situations) have not been much
documented in the automobile industry. Therefore we feel that single source robustness should be studied in depth prior
to more general cases.

[Meta-Review · NeurIPS 2019]

This is a borderline paper that discusses the problem of robustness with two sources. The reviewers thought the approach was novel; however, there is a lack of theoretical analysis and question about the method's generalization to more complex scenarios.